# A Handy Flexible Micro-Thermocouple Using Low-Melting-Point Metal Alloys

**DOI:** 10.3390/s19020314

**Published:** 2019-01-14

**Authors:** Qifu Wang, Meng Gao, Lunjia Zhang, Zhongshan Deng, Lin Gui

**Affiliations:** 1CAS Key Laboratory of Cryogenics, Technical Institute of Physics and Chemistry, Chinese Academy of Sciences, Beijing 100190, China; wangqifu16@mails.ucas.ac.cn (Q.W.); mgao@mail.ipc.ac.cn (M.G.); zhanglunjia14@mails.ucas.ac.cn (L.Z.); zsdeng@mail.ipc.ac.cn (Z.D.); 2School of Future Technology, University of Chinese Academy of Sciences, Beijing 100149, China

**Keywords:** micro-thermocouple, flexible sensor, low-melting-point alloys, liquid metal

## Abstract

A handy, flexible micro-thermocouple using low-melting-point metal alloys is proposed in this paper. The thermocouple has the advantages of simple fabrication and convenient integration. Bismuth/gallium-based mixed alloys are used as thermocouple materials. To precisely inject the metal alloys to the location of the sensing area, a micro-polydimethylsiloxane post is designed within the sensing area to prevent outflow of the metal alloy to another thermocouple pole during the metal-alloy injection. Experimental results showed that the Seebeck coefficient of this thermocouple reached −10.54 μV/K, which was much higher than the previously reported 0.1 μV/K. The thermocouple was also be bent at 90° more than 200 times without any damage when the mass ratio of the bismuth-based alloy was <60% in the metal-alloy mixture. This technology mitigated the difficulty of depositing traditional thin–film thermocouples on soft substrates. Therefore, the thermocouple demonstrated its potential for use in microfluidic chips, which are usually flexible devices.

## 1. Introduction

Temperature is one of the most important physical signals in many microfluidic applications. For example, the rate of temperature change in the sub-microliter-scale polymerase chain reaction (PCR) directly affects the efficiency of DNA extraction and separation [1,2]. Temperature gradient can help to separate ionic species in a microchannel or a capillary device [3]. Variations in temperature in a microreactor illustrate the process of chemical reaction [4]. A single cell may have a gene mutation because of a temperature change [5].

Thermocouples [6,7], thermal resistance sensors [8], and optical methods [9,10] are traditionally usually used to measure microfluidic temperature. Optical systems can measure the entire microfluidic channel when thermo-photosensitive reagents are dispersed throughout the sample fluid. However, these sensitive reagents contaminate the sample fluid.

Solid-metal thin films are widely used to fabricate thermocouples or thermal-resistance sensors by deposition or sputtering [11]. These thin-film sensors have high resolution and quick response because of their small size (usually nanometer-level thickness). However, thin-film sensors have some limitations. For example, the process of deposition or sputtering solid metal onto a rigid substrate is complicated, expensive, and time-consuming. Solid-metal thin films also usually have a coefficient of thermal expansion (CTE) different from that of the substrate. Consequently, thin-film sensors have been easily damaged when temperatures rise [8]. Moreover, they have been proven unsuitable for some microfluidic systems requiring flexibility and deformation. With the development of microfluidics, a demand has emerged to develop a flexible temperature microsensor with simple fabrication.

Low-melting-point metals or alloys (gallium, gallium-based alloys, and bismuth-based alloys) have been used in microfluidic systems. An important merit of these metals or alloys is that they remain liquid around room temperature. They can be easily injected into microchannels to fabricate microdevices using a handy syringe [12]. In particular, these microdevices can work well even if they are heated, bended, or twisted [13]. In 2007, for the first time, Siegel et al. [14] proposed a method of fabricating metallic microstructures in microchannels by injecting liquid solder at 180 °C. Recently, a room-temperature, gallium-based-alloy liquid metal was used to fabricate many devices in microscale channels, such as an electroosmotic flow pump [15], sensors [16,17], antennas [18], and electrodes [19]. Dickey et al. [12] used liquid metal to fabricate a stable microstructure in a microchannel in 2008. Gao et al. [20] proposed a liquid-metal resistance temperature detector to measure the microscale temperature from 20 °C to 70 °C. They also demonstrated a fast-thermal response microfluidic system using liquid metal in 2016 [21]. However, the shape of the liquid–metal microchannel has an important impact on the thermal resistance of the sensor. The resistance of the liquid-metal microstructure changes largely when the sensor is bent or twisted [16]. Conversely, thermocouples do not have this shortcoming because the Seebeck coefficient of their material is irrelevant to their shape and size. 

However, the Seebeck coefficient of thermocouples fabricated using room-temperature gallium-based-alloy liquid metal is very small (0.1 μV/K) and cannot be measured easily using normal equipment [20,22]. Bismuth has a much higher Seebeck coefficient (−75.4 μV/K) than indium (6.9 μV/K), tin (4.2 μV/K), and gallium (0.1 μV/K) with a platinum electrode as the connection wire [20,23]. Thus, in the present work, we present a bismuth-based alloy combined with a room-temperature gallium-based-alloy liquid metal to fabricate a flexible thermocouple with a high Seebeck coefficient.

## 2. Materials and Methods

### 2.1. Preparation of Metal Alloys

Room temperature liquid-metal eutectic gallium–indium–tin alloy (EGaInSn; 68.5% Ga 21.5% In, 13% Sn by weight; 11 °C melting point) was used to fabricate one pole of the thermocouple. Two low-melting-point Bi-based metal–alloy mixtures, namely, EBiIn–EGaInSn and EBiInSn–EGaInSn, were used to fabricate the other pole of the thermocouple. EBiIn–EGaInSn is a mixture of eutectic bismuth–indium (EBiIn; 67% Bi 33% In by weight; 109 °C melting point) and eutectic gallium–indium–tin (EGaInSn; 68.5% Ga 21.5% In 13% Sn by weight; 11 °C melting point). EBiInSn–EGaInSn is a mixture of eutectic bismuth–indium–tin (EBiInSn; 32.5% Bi 51% In 15.5% Sn by weight; 60 °C melting point) and eutectic gallium–indium–tin (EGaInSn; 68.5% Ga 21.5% In 13% Sn by weight; 11 °C melting point). 

To prepare these two mixtures, a fixed mass of solid EBiIn or a fixed mass of EBiInSn was initially placed on a thermostatic plate at 125 °C. When the solid EBiIn or EBiInSn melted, a fixed mass of EGiInSn was added into EBiIn or EBiInSn. After stirring the mixture with a glass rod for 3 min, an EBiIn–EGaInSn mixture was obtained. Six mixtures of EBiIn–EGaInSn or EBiInSn–EGaInSn (40%, 50%, 60%, 70%, 80%, and 90% mass ratio of EBiIn or EBiInSn in total mixture) were prepared to determine the optimal mixture for fabricating the thermocouples.

### 2.2. Chip Design and Fabrication

Figure 1a shows the microfluidic chip embedded with a handy, flexible micro-thermocouple using low-melting-point metal alloys. This microfluidic chip consisted of a polydimethylsiloxane (PDMS) slab (1 mm thick) and a PDMS membrane (50 μm thick). The PDMS slab had an ohm-shape microchannel (5 cm long, 100 μm wide, and 50 μm high) for the thermocouple. Two identical injection holes were designed 2.5 mm away from the microchannel. One pole of the thermocouple microchannel was filled with EGaInSn, and the other was filled with the Bi-based metal–alloy mixture. The PDMS gap between these two thermocouple poles was 100 μm. 

A challenge for injecting these two different metal alloys into the microchannel to fabricate thermocouples was that both metal alloys had to flow smoothly in the microchannel and accurately stop at the sensing area. When injecting one of these two metal alloys into its thermocouple pole, overflowing to another thermocouple pole was easy, leading to thermocouple fabrication failure. To prevent this problem, we designed a micro-PDMS post within the sensing area to increase the flow resistance of metal alloy and serve as a control point for the injection of metal alloy. As shown in Figure 1b, the micro-PDMS post (50 μm long and 30 μm wide) was fabricated at the intersection of these two thermocouple poles to ensure that the EGaInSn and the Bi-based metal–alloy mixture converged well at the micro-PDMS post. To vent the air from the microchannel when injecting metal alloys, a 30 μm-wide small microchannel was fabricated near the sensing area (Figure 1b).

All chips for thermocouples were fabricated by standard soft-lithography. SU-8 2050 (MicroChem Corp., Worcester, MA, USA) was used to fabricate 50 μm-high microchannel patterns for thermocouple poles on a 4-in silicon wafer. Liquid PDMS (PDMS monomer and a curing agent at a 10:1 ratio by weight, Dow Corning Corp., Wiesbaden, Germany) was directly poured onto the pattern to fabricate the PDMS slab containing thermocouple poles. Then, the PDMS slab was peeled off from the wafer after being cured at 60 °C for 2.5 h. To obtain the PDMS membrane (50 μm thick), liquid PDMS was spin-coated on a silicon wafer (speed, 1500 rpm; spin time, 30 s) and then baked at 75 °C for 30 min. The PDMS slab and the PDMS membrane were irreversibly bonded together using oxygen plasma treatment (plasma cleaner, YZD08-2C, Tangshan Yanzhao Technology, Tangshan, China).

The EGaInSn and the Bi-based metal–alloy mixture were injected into the microchannels using a syringe. The mixture was initially melted on a thermostatic plate at 125 °C, and then filled into the pole of the microchannel from the injection hole using an electrical heating syringe (110 °C). By adjusting the injection pressure and time, the mixture flowed smoothly into the microchannel and stopped when it came into contact with the micro-PDMS post. After cooling this mixture to room temperature in the microchannel, the EGaInSn was then injected into different pole of the microchannel than the injection hole. To keep full contact with the mixture around the micro-PDMS post, EGaInSn was pumped into the microchannel for a much longer injection time until it overflowed from the small microchannel near the sensing area (Figure 1c). Finally, copper-leading wires (200 μm diameter) were inserted into the two injection holes, which were sealed with a package adhesive sealant (705 RTV Transparent Silicone Rubber). The outflow reservoir of the small microchannel was also sealed using this adhesive sealant.

### 2.3. DSC Measurement

To understand the physical properties of the low-melting-point Bi-based metal–alloy mixture, differential scanning calorimetry (DSC) was performed. The rates of temperature-up and temperature-down were 10 °C/min.

### 2.4. Electron Microscope Observation

To analyze the morphology features of these metal–alloy mixtures, these materials were observed in the electron microscope. The shape, size, and stoichiometry of the materials were obtained.

### 2.5. Calibration

To calibrate the fabricated thermocouples, the sensing area of each thermocouple was placed onto a thermostatic plate, and the other part of the thermocouple with copper-leading wires was fully immersed in an ice–water mixture. Standard K-type thermocouples were used to measure the temperature of the thermostatic plate and the ice–water mixture. The thermoelectric potential was monitored with an Agilent 34970A (Keysight Technologies, Santa Rosa, CA, USA). During calibration, the temperature of the thermostatic plate increased by 5 °C after each test and was maintained for 3 min. The data acquisition frequency was 2 Hz.

### 2.6. Bending Test

To test the bending performance of these thermocouples, they were suspended on a flat panel. The thermocouples were bent 200 times at 90° bending angle (Figure 1d).

### 2.7. Temperature Measurement of Microheater

To test the performance of the thermocouple in a microscale temperature measurement, a microfluidic chip with an EGaInSn liquid-metal microheater was fabricated. As shown in Figure 1e, the channel width of the heating area in the microheater was 100 μm and the channel width of the extended section was 200 μm. The resistance of the microheater was 3.6 Ω. The thermocouple was placed over the microheater. A 20 μm-thick PDMS membrane was placed between the thermocouple and the microheater (Figure 1e). A DC power supply (DH1720A, DaHua Power, resolution 15 mA) was used to apply heating voltages to the microheater.

## 3. Results and Discussion

Low-melting-point Bi-based alloys as the material of thermocouples have higher Seebeck coefficient than room-temperature Ga-based alloys. However, Bi-based alloys are easily damaged after filling and solidifying into microchannels because of their brittleness [19]. To prevent this problem, mixtures with Bi-based alloy and EGaInSn were proposed in this work. 

Figure 2 shows the images of six EBiIn–EGaInSn alloys and six EBiInSn–EGaInSn alloys with different mass ratios of Bi-based alloy in mixtures. As shown in Figure 2ai,bi, the mixture with 40% Bi-based alloy and 60% EGalnSn was liquid, with some solid particles, at room temperature. When the mass ratio of Bi-based alloy reached 60%, the mixture remained sticky but still was not fully solidified (Figure 2a(iii),b(iii)). When the mass ratio of the Bi-based alloy was close to or more than 70%, the mixture became a powdery solid after cooling (Figure 2a(iv–vi),b(iv–vi)). The mixture with less than 40% mass ratio of Bi-based alloy was not considered because it did not have sufficient solid particles filled in the microchannel, which led to the easy movement of solid particles within the liquid metal. As a result, the mixture easily back-flowed from the channel when EGaInSn was injected from the other pole of the channel. The 100% mass ratio of Bi-based alloy was also not considered because of its brittleness.

As shown in Figure 3, all the mixtures were scanned in the scanning electron microscope (SEM). Figure 3a,b show the SEM images for the textures of different ratio of bismuth/gallium-based alloy mixtures. In order to reveal the distribution of metal elements, all of these textures were examined through energy dispersive spectrum and Figure 3c shows the results from the alloy mixture with 80% Bi-based alloy. Comparing Figure 3c(ii,cv) or Figure 3c(vii,cx), it is found that the distribution of Ga and Bi is not uniform, and they seem to “repel” each other, which indicates that Ga and Bi do not mix in microscale. Figure 3c(iii) or Figure 3d shows the distribution map of In in 80% EBiInSn. As shown in Figure 3d, the red dashes and yellow dashes represent the area of Bi and Ga, respectively. It can be seen that the distribution of In in the red (Bi) area was much heavier than that in the yellow (Ga) area, which means that the In in the Bi-based alloy did not diffuse into Ga-based alloy. On the contrary, as shown in Figure 3c(iv,cix), Sn distributed almost uniformly throughout the whole area, except in the empty space SEM cannot detect. What should be pointed out is that, in Figure 3c(ix), the Sn was supposed to be absent in the Bi area because EBiIn did not contain any Sn before the mixing. The Sn was still quite uniformly distributed in the whole area, both in Ga–area and Bi–area, which means the Sn entered EBiIn very easily during the mixing. In summary, the Ga-based alloy and Bi-based alloy are thoroughly intermingled with each other and filled the entire space (SEM images for the mixture with other ratios can be found in the Appendix A).

To identify the specific ingredients of the mixtures, we performed DSC tests on all mixtures. Figure 4a shows the DSC phase diagrams of the Bi-based alloy mixture with 50% mass ratio of EBiInSn. Four peaks indicate that the mixture comprised two metal alloys with different melting points. The first and fourth peaks represented the melting endothermic process and the solidification exothermic process of one metal alloy in the mixture, respectively. Figure 4a clearly shows that the melting point was 11.06 °C and the solidification temperature was –42.61 °C. This melting point was equal to the melting point of EGaInSn. Thus, the liquid portion of the mixture was EGaInSn at room temperature. Similarly, the melting and freezing points of the second metal were 42.00 and 32.72 °C. The melting point of this metal was lower than that of EBiInSn (60 °C). This metal may be a new metal–alloy mixture of EBiInSn and EGaInSn. The formation of this new metal can be explained from the energy spectrum. As mentioned above, in the energy spectrum analysis, during the mixing, Sn transferred between Ga-based alloy and Bi-based alloy easily. The Sn transfer phenomena may generate this new alloy.

As shown in Figure 4b, the liquid portion of the mixture was EGaInSn, and the solid portion was the new alloy. Comparing the other phase diagrams with different mass ratio of Bi-based alloy (see Appendix A) with the increment of the portion of Bi-based alloy, the peak of EGaInSn decreased and the new metal alloy increased. It can also be seen that the liquid portion of the mixture decreased and the solid portion increased with the mass ratio increment of Bi-based alloy.

Figure 5a,b show the thermoelectric potential of the thermocouples with different mass ratios of Bi-based alloy at different temperatures. The thermoelectric potential of the thermocouple increased with increased mass ratio of Bi-based alloy at a fixed temperature. With further increased temperature of the thermocouple, the increase rate of the thermoelectric potential at a fixed temperature decreased abruptly. This fixed temperature was higher than the melting point of the mixture. The thermoelectric potential almost linearly increased with increased temperature before the mixture melted. Thus, the thermocouple measured microfluidic temperature below the melting point of the mixture.

Moreover, when the temperature was higher than the melting point of mixture, the thermoelectric potential did not change any further. Although the mixture at the hot end completely melted, an incomplete solidified mixture remined at the cold end. The maximum temperature of this part of the mixture was close to the melting temperature. Thus, this part of the unsolidified mixture was responsible for maintaining the thermoelectric constant. 

The Seebeck coefficient is an important factor in evaluating the performance of thermocouples. The thermoelectric potential of the thermocouple is related to the material properties and the structure of the thermocouple, according to the theoretical formula of a thermocouple as follows:(1)$$E_{AB}(T,T_{0})=\alpha_{AB}\cdot \left(T_{h}-T_{c}\right),$$
where $E_{AB}$ is the thermoelectric power of the thermocouple, $\alpha_{AB}$ is the Seebeck coefficient of thermocouple wire material, and $T_{h}$ and $T_{c}$ are the temperature at the hot and cold junctions, respectively. We obtained the Seebeck coefficient of the thermocouple from Equation (1). 

Figure 5c shows the Seebeck coefficients of the EBiIn–EGaInSn thermocouple and the EBiInSn–EGaInSn thermocouple. A higher mass ratio of bismuth meant a higher Seebeck coefficient. The Seebeck coefficient of EBiIn–EGaInSn was also higher than that of EBiInSn–EGaInSn. Among all of these thermocouples, the highest and lowest Seebeck coefficients were −0.55 and −10.54 μV/K, respectively, which were much greater than the previously reported 0.1 μV/K [20]. However, a high ratio of Bi-based alloy in mixture corresponded to a high Seebeck coefficient of the thermocouples, but they became easily damaged when bent. Figure 5d shows the results of the bending test of these thermocouples. Thermocouples with 40%, 50%, and 60% mass ratios of the Bi-based alloy in mixture still worked well after being bent 200 times continuously. Thermocouples with 70%, 80%, and 90% mass ratio of the EBiIn in the mixture did not present flexibility and were damaged during the first bending test. Thermocouples with 70% and 80% mass ratios of EBiInSn were damaged after bending tests were done, five and two times, respectively. The thermocouple with 90% mass ratio of EBiInSn was also damaged after the first bending test. The decrease in mass ratio of the Bi-based alloy in the mixture improved the flexibility of the thermocouples. In other words, because of the presence of the room–temperature liquid–metal EGaInSn in the mixture, the thermocouple eliminated the brittleness of bismuth and was thus flexible and bendable. In microfluidic systems, when a thermal microchip requires thermocouples with high flexibility, the mass ratio of Bi-based alloy in the mixture for the thermocouple can be suggested to be <60%. After the bending test, the undamaged thermocouple was tested again. Compared to its state before bending, the thermocouple’s Seebeck coefficient was nearly equal (see Appendix A). 

In addition, to test the stability of these thermocouples, each thermocouple was tested for three heating and cooling cycles. The results showed that the Seebeck coefficients of these thermocouples were basically unchanged in the effective temperature range (see Appendix A). It is noteworthy that the alloy mixture was definitely not as stable as pure solid material, however, we still suggest that, for high accuracy, a calibration be required before using these “soft” thermocouples.

The thermocouple with 40% mass ratio of EBiIn was chosen to measure the temperature of EGaInSn microheater. Using the calibration curve (Figure 5a), the voltage signal collected by the liquid-metal thermocouple was converted into a temperature signal. The test was performed at 25 °C. Figure 6 shows the experimental results of temperature measurement for the liquid-metal microheater. With increased voltage by 0.5 V per test, the thermocouple monitored the temperature change over time. When the voltage reached 0.65 V, the temperature of the microheater was 83 °C. These findings indicated that this thermocouple worked well below 83 °C.

## 4. Conclusions

A low-melting-point Bi-based alloy micro-thermocouple was proposed and fabricated using fast microfluidic injection method for the first time. This technology mitigated the difficulty of depositing traditional thin-film thermocouples on soft substrates. By mixing a proper amount of EGaInSn alloy into the Bi-based alloy, the thermocouple can present good flexibility and deformability. Moreover, the results of the SEM illustrated that the Ga-based alloy and Bi-based alloy were thoroughly intermingled with each other and filled the entire channel. The thermocouple had a significant thermoelectric potential that reached –10.54 μV/K, much larger than the previously reported 0.1μV/K. After three repeated heating/cooling processes, these thermocouples still maintained good linearity. 

This kind of thermocouple could easily be integrated into microfluidic systems, if the structures of both thermocouple and microfluidic system are simultaneously designed and fabricated in just one step of soft lithography. Thus, this thermocouple proved its potential for use in many thermal microfluidic applications (which is also our future work), such as PCR chip, on-chip cell culture, etc. In addition, it was found that the distribution of each of the four elements—Ga, Bi, In, and Sn—was quite different. Particularly, Sn was almost uniformly distributed in the whole space, while others were not. Based on this knowledge, it was speculated that Sn had a greater influence on the uniformity and stability of the mixtures. Thus, another future work might be analyzing the relationship between the Sn and the thermoelectric effect of the mixtures.

## Figures and Tables

**Figure 1 sensors-19-00314-f001:**
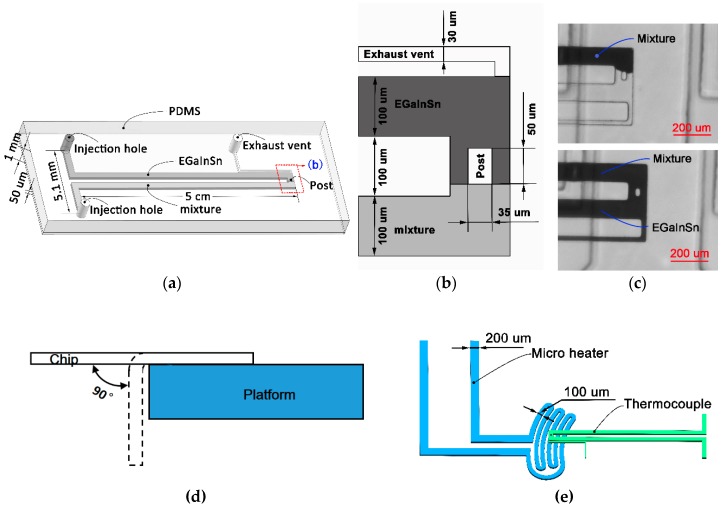
(**a**) Schematic of the ohm-shape micro-thermocouple; one pole of the channel was filled with EGaInSn, and the other pole of the thermocouple was filled with low-melting-point Bi-based metal–alloy mixture. (**b**) Zoom in of schematic (**a**), wherein EGaInSn and the mixture converged at the middle of the channel (sensing area). (**c**) Local map of the thermocouple chip showing that one pole of the thermocouple channel was filled with the mixture and another pole was filled with EGaInSn. (**d**) Sketch of bending test. (**e**) Sketch of temperature-measurement test. A 20 μm-thick polydimethylsiloxane (PDMS) membrane was placed between the thermocouple and the microheater chip.

**Figure 2 sensors-19-00314-f002:**
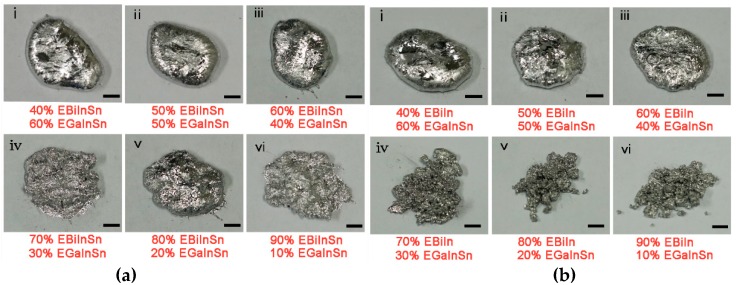
(**a**) Image of different mass ratios of EBiInSn in mixture. (**b**) Image of different mass ratios of EBiIn in mixture. With increased mass ratio of bismuth-based alloy, the mixture changes from a liquid to a powdery solid. Scale bars: 1 mm (the experimental mixing Appendix A are included in the Appendix A).

**Figure 3 sensors-19-00314-f003:**
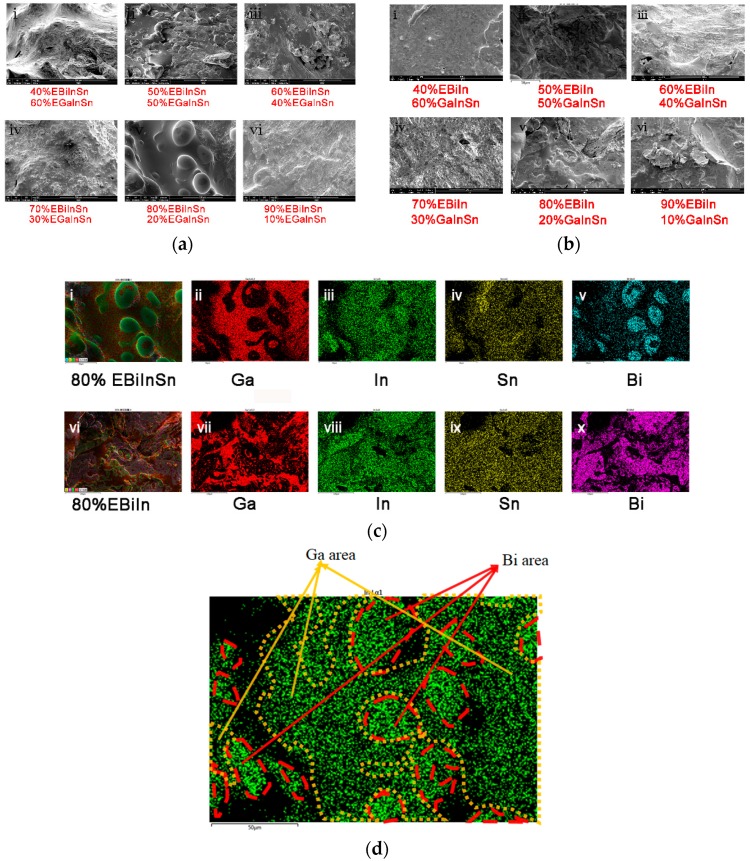
(**a**) SEM graphics of EBiInSn-based alloy mixtures. (**b**) SEM graphics of EBiIn-based alloy mixtures. (**c**) Energy spectrum diagram of 80% Bi-based alloy mixtures. The element gallium and bismuth were intermingled with each other, filling the entire space. (**d**) Distribution map of In in 80% EBiInSn. The area in red dashes and the yellow dashes represent the area of Bi and Ga, respectively.

**Figure 4 sensors-19-00314-f004:**
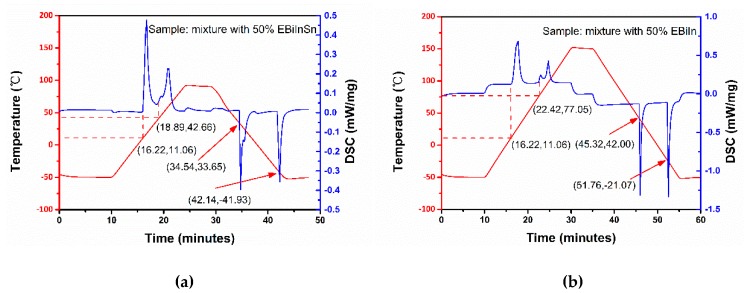
(**a**) Differential scanning calorimetry (DSC) phase diagram of mixture with 50% mass ratio of EBiInSn. (**b**) DSC phase diagram of mixture with 50% mass ratio of EBiIn.

**Figure 5 sensors-19-00314-f005:**
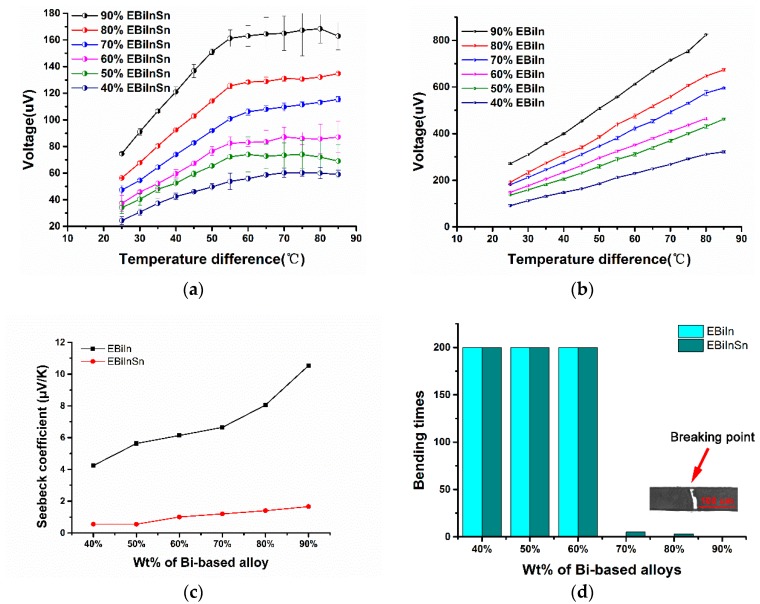
(**a**) Calibration curve of thermocouples with different ratios of EBiInSn. (**b**) Calibration curve of thermocouples with different ratios of EBiIn. (**c**) Seebeck coefficients of thermocouples calculated by the data of the calibration curve. (**d**) Results of bending test of thermocouples with different ratios of Bi-based alloy (the experimental bending test Appendix A are included in the Appendix A).

**Figure 6 sensors-19-00314-f006:**
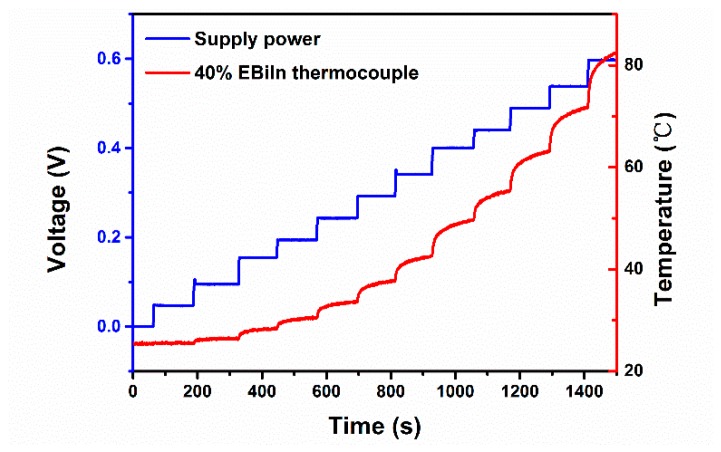
Experimental results of the performance test of thermocouple. The thermocouple with 40% ratio of EBiIn was used to monitor the temperature change of GaInSn microheater.

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
