# Peer review of "A Handy Flexible Micro-Thermocouple Using Low-Melting-Point Metal Alloys"

_sensors, 2019, doi:10.3390/s19020314_

Round 1
Reviewer 1 Report
In this work, authors have presented a flexible micro thermocouple for measuring temperature in microfluidic platforms.The work presented is interesting, however there are some issues that need to be addressed.
Authors should consider re-structuring the abstract to stress on significance of a micro-thermocouple and how their work affects this application.
In Introduction, authors refer to microfluidic detection. This term is very generic. Please elaborate with specific examples and references.
Authors need to thoroughly proof read their document as there are many grammatical errors which impact the effectiveness of the manuscript.
Introduction, line 57: Please provide a reference for this statement. How have authors addressed this in their work?; especially if they claim that the shape of the microchannel is significant.
Introduction lacks background information and does not elaborate on significance of temperature measurements in microfluidic applications.
How do the authors propose to integrate their thermocouple with existing bio-microfluidic platforms?
None of the plots presented have error bars.
What is the repeatability and precision of measurement?
Author need to expand their discussion and conclusion sections to describe the significance of their work and compare with existing technologies,
Author Response
Response to reviewers’ comments:
In this work, authors have presented a flexible micro thermocouple for measuring temperature in microfluidic platforms. The work presented is interesting, however there are some issues that need to be addressed.
Authors should consider re-structuring the abstract to stress on significance of a micro-thermocouple and how their work affects this application.
Reply: Thanks for the reviewer’s comments. We have re-built our abstract. We highlighted that these thermocouples have advantages of easy fabrication comparing to traditional micro thin-film thermocouples. For traditional thin-film thermocouples, the substrate needs to be rigid for the metal sputtering process. The thermocouple proposed in this work can be easily integrated in a microfluidic chip, especially for soft substrate like PDMS. We have add these description in the revised abstract.
In Introduction, authors refer to microfluidic detection. This term is very generic. Please elaborate with specific examples and references.
Reply: Thanks for the reviewer’s comments. We have specified the description of microfluidic detection. We provided a reference about an example that monitoring the temperature of PCR solution in the microchannel. All the modifications have been done in our revised manuscript (See Page 1, line 27 to 28).
Authors need to thoroughly proof read their document as there are many grammatical errors which impact the effectiveness of the manuscript.
Reply: Thanks for the reviewer’s suggestions. We have modified the manuscript carefully.
Introduction, line 57: Please provide a reference for this statement. How have authors addressed this in their work?; especially if they claim that the shape of the microchannel is significant.
Reply: Thanks for the reviewer’s suggestions. We have found reference about an application that illustrates this property. This is a resistance sensor. When channel was squeezed, the space of the channel would become smaller. Then the resistance of the liquid metal would change. This reference has been added in revised manuscript (See Page 2, second paragraph, line 58)
The ID of this reference is Ref.16: “DOI: 10.3390/s150511823”.
Introduction lacks background information and does not elaborate on significance of temperature measurements in microfluidic applications.
Reply: Thanks for the reviewer’s suggestions. We highly agree with the reviewer’s suggestion. We have added more background on microfluidic temperature measurements in the introduction. Four applications in the microfluidic temperature measurement have been added in the revised introduciton to show the significance of temperature measurements. All the modifications have been added in the revised manuscript. (See Page 1, line 27 to 31)
How do the authors propose to integrate their thermocouple with existing bio-microfluidic platforms?
Reply: Thanks for the reviewer’s comments. This thermocouple can be easily integrated in microfluidic system when the structures of thermocouple and microfluidic system are fabricated in one substrate through soft lithography. The only step needs to do is to inject the liquid mixtures into the micro channel to form the micro-thermocouples. All these contents have been added in the revised manuscript (Page 9, the “Conclusions” part, Line 311 to 314).
None of the plots presented have error bars.
Reply: Thanks for the reviewer’s comments. It was our mistake. We have added error bars for all the plots in the revised manuscript.
What is the repeatability and precision of measurement?
Reply: Thanks for the reviewer’s comments. We have added some new experimental results in our revised paper and supplementary documents to show the repeatability and precision. We put all the thermocouple in three heating/cooling cycles and find that after the heating cycles the Seebeck coefficients are almost same (see part 3 in supplementary document). We also compared the performance of thermocouple before the bending test with the performance after bending test. The performance is almost same and highly repeatable (see part 4 in supplementary document). The error bars added in the plots show that the precision is in an acceptable range for normal temperature measurement. (Page 8, line 282).
Reviewer 2 Report
The authors present the novel construction of a simultaneously flexible and highly sensitive thermocouple based upon liquid metal intelligent rationale of utilizing various low melting point liquid metal alloys. To my knowledge this paper presents the first attempt of utilizing both bismuth based and gallium based to achieve these properties. The authors present rational for utilizing bismuth/gallium based mixed alloys as Seebeck elements and analyse the metallurgic and physical properties of the alloys. Furthermore, the authors demonstrate the successful sensing capabilities of their devices and resilience to mechanical stresses (bending).
However, there are several fundamental gaps in the understanding of the underlying mechanisms which are cause for concern. Clarification of these would greatly improve the manuscript.
1. Figure 2 presents physical images of the alloy mixtures (and the supporting videos do likewise) however, there is little analysis of the size/shape/stoichiometry of the solid particles within the liquid alloys. Further morphological and metallurgic analysis would reveal a greater understanding of the system.
2. Line 216/equation 1 present a simplified and generalized thermodynamic understanding of the Seebeck effect at junction. This determines the Seebeck coefficient of the complete device, however for scientific understanding may not be truly applicable as the thermodynamic framework that derived this relationship possibly assumes homogeny within each material (There is no reference given for this relation).
a. Furthermore the Seebeck coefficient may change as the consistency of the material changes (as the authors point out while discussing figure 4).
3. Figure 3 presents DSC of 50% EBiIn(Sn)/EGaInSn and established the melting and freezing points thereof. I have several concerns
a. 40%, 60%, 80% and 90% are not presented although the manuscript mentions that DSC was performed on such alloys. Do the other mixtures behave differently? If so then this should be presented in some form (perhaps as supplementary information) and is not then this should be discussed in the manuscript.
b. It is mentioned in line 189 that the authors presume that the EBiIn(Sn)/EGaInSn mixture forms a new separate alloy. Does this change with multiple heating/cooling cycles? Would this have an effect the sensing capabilities?
c. In Figure 4 (a & b) the boundary between linear/nonlinear response is marked. The authors rationalize this as the melting point of the Bi alloy although this is different from that observed in the DSC. Furthermore the linear/nonlinear boundary appears to differ with mixture %. Authors would be advised to compare the DSC for each mixture to the linear limit of the device.
4. Did the authors consider the reproducibility of the sensing (Figure 4)? This ties back into the concerns with the DSC,
a. After multiple heating/cooling cycles do the Seebeck coefficients change? If so do they stabilize after a certain number of cycles?
b. Furthermore, on line 170 the authors mention potential ‘backflow’ of the EBiIn(Sn) into the EGaInSn for mixtures below 40%. Does this affect high temperature measurements for other mixtures? For example is it possible for the 40% EBiInSn mixture to mix with the EGaInSn at high temperatures? If so how does this effect the performance?
5. Were the thermocouples responses characterized after the bending test? A comparison of before, during and after bending would highlight the resilience of the device.
6. Were other matching materials considered? For example EGaIn which is also liquid at room temperature. If so why specifically was EGaInSn chosen?
Author Response
Response to reviewers’ comments:
The authors present the novel construction of a simultaneously flexible and highly sensitive thermocouple based upon liquid metal intelligent rationale of utilizing various low melting point liquid metal alloys. To my knowledge this paper presents the first attempt of utilizing both bismuth based and gallium based to achieve these properties. The authors present rational for utilizing bismuth/gallium based mixed alloys as Seebeck elements and analyse the metallurgic and physical properties of the alloys. Furthermore, the authors demonstrate the successful sensing capabilities of their devices and resilience to mechanical stresses (bending).
However, there are several fundamental gaps in the understanding of the underlying mechanisms which are cause for concern. Clarification of these would greatly improve the manuscript.
Figure 2 presents physical images of the alloy mixtures (and the supporting videos do likewise) however, there is little analysis of the size/shape/stoichiometry of the solid particles within the liquid alloys. Further morphological and metallurgic analysis would reveal a greater understanding of the system.
Reply: Thanks for the reviewer’s suggestions. According to reviewer’s suggestions, all the gallium/bismuth based alloy mixtures with different mixing ratio were scanned in the scanning electron microscope. The SEM images help us a lot to understand the alloy mixtures. We found that the Ga and Bi do not mix in microscale and In does not transfer between Ga-based alloy and Bi-based alloy. Surprisingly, Sn transfers quite easily between these two alloys. We have added these new findings in the revised manuscript (See Page 5, second paragraph, line 183 to 210). All the SEM images can be found in chapter 1 of supplementary document.
Line 216/equation 1 present a simplified and generalized thermodynamic understanding of the Seebeck effect at junction. This determines the Seebeck coefficient of the complete device, however for scientific understanding may not be truly applicable as the thermodynamic framework that derived this relationship possibly assumes homogeny within each material (There is no reference given for this relation).
a. Furthermore the Seebeck coefficient may change as the consistency of the material changes (as the authors point out while discussing figure 4).
Reply: Thanks for the reviewer’s comments. We highly agree with reviewer’s comments. Equation 1 is simplified and based on the assumption that the Seebeck coefficient of all the alloys does not change with temperature. According to our experimental results in figure 5(a) and (b), the thermoelectrical potential is quite linear to the temperature difference before any phase change happens in the Bi-based alloy mixture (25-83°C for EBiIn-EGaInSn and 25-55°C for EBiInSn-EGaInSn). So, no mater how equation 1 was derived, this equation is correct to depict the relationship between the temperature difference and thermoelectrical potential based on the experimental results. In fact, to some degree, the typical Cu-NiGu T-type thermocouple is also like our “Ga-BiGa” thermocouple. The difference is that we replaced Cu with Ga-based alloy and Ni with Bi-based alloy. According to our SEM results, the Ga-based alloy and Bi-based alloy mixed with each other quite thoroughly in our experiments. Because the alloy mixture are definitely not as stable as pure solid material, we still suggest that before using these “soft” thermocouples, a calibration is necessary if high accuracy is required. We have mentioned this in our revised manuscript (See Page 8, second paragraph, line 281 to 282).
3. Figure 3 presents DSC of 50% EBiIn(Sn)/EGaInSn and established the melting and freezing points thereof. I have several concerns
a. 40%, 60%, 80% and 90% are not presented although the manuscript mentions that DSC was performed on such alloys. Do the other mixtures behave differently? If so then this should be presented in some form (perhaps as supplementary information) and is not then this should be discussed in the manuscript.
Reply: Thanks for the reviewer’s comments. We did the DSC test in order to understand the ingredients of the alloy mixtures. According to reviewer’s suggestion, we have added all the DSC results in the supplementary material. As shown in chapter 2 of supplementary document, other mixtures behave quite same. Thanks for the reviewer’s suggestion, the revised manuscript and supplementary information are more informative.
b. It is mentioned in line 189 that the authors presume that the EBiIn(Sn)/EGaInSn mixture forms a new separate alloy. Does this change with multiple heating/cooling cycles? Would this have an effect the sensing capabilities?
Reply: Thanks for the reviewer’s suggestions. Reviewer’s suggestion is quite helpful. According to reviewer’s suggestion, we did heating/cooling cycles for three times for each type of thermocouple. Figure 5 (a) and (b) show the experimental results. The results show that the performance of these thermocouples is relatively stable in their working range which are below 80°C and 55°C, respectively. All these contents have been added in the revised manuscript (See Page 8, second paragraph, line 278 to 281) and in chapter 3 of supplementary materials.
c. In Figure 4 (a & b) the boundary between linear/nonlinear response is marked. The authors rationalize this as the melting point of the Bi alloy although this is different from that observed in the DSC. Furthermore the linear/nonlinear boundary appears to differ with mixture %. Authors would be advised to compare the DSC for each mixture to the linear limit of the device.
Reply: Thanks for the reviewer’s suggestions. We did all the DSC tests. We found that the working range (<80°C and <55°C) are a little bit higher than the melting point of the new solid metal alloys which is around 73°C and 43°C, respectively. The difference is about 7~12°C. Unfortunately, we have not find very proper and solid reason to explain this difference. One of the possible reasons might be the size effect, because the size of the micro thermocouple is about 100micron and the sample for DSC is 1millimeter in size.
4. Did the authors consider the reproducibility of the sensing (Figure 4)? This ties back into the concerns with the DSC,
a. After multiple heating/cooling cycles do the Seebeck coefficients change? If so do they stabilize after a certain number of cycles?
Reply: Thanks for the reviewer’s suggestions. We did these heating/cooling cycles for three times. The Seebeck coefficients are almost same in the working range. We have mentioned these tests in the revised manuscript (See Page 8, second paragraph, line 278 to 280). All the experimental data have been added in the supplementary document.
b. Furthermore, on line 170 the authors mention potential ‘backflow’ of the EBiIn(Sn) into the EGaInSn for mixtures below 40%. Does this affect high temperature measurements for other mixtures? For example is it possible for the 40% EBiInSn mixture to mix with the EGaInSn at high temperatures? If so how does this effect the performance?
Reply: Thanks for the reviewer’s comments. Yes, this will affect high temperature measurements for all other mixtures. It is possible that mixed liquid alloys will enter the other channel. The thermocouple would not work if both channels have same mixed alloys based on the principle of Seebeck effect. In order to avoid this problem, we suggest that the working range of these thermocouples are lower than their melting points. According to the experimental results, when the temperature gets too high, the voltage will become unstable.
5. Were the thermocouples responses characterized after the bending test? A comparison of before, during and after bending would highlight the resilience of the device.
Reply: Thanks for the reviewer’s suggestions. We highly agree with reviewer’s suggestion and did all the tests according to reviewer’s suggestion. The results show that all the flexible thermocouples that with 40%-60% Bi-based alloys. The Seebeck coefficients are almost same in the working rang. We have added the experimental data in chapter 4 of supplementary document.
6. Were other matching materials considered? For example EGaIn which is also liquid at room temperature. If so why specifically was EGaInSn chosen?
Reply: Thanks for the reviewer’s comments. In the manuscript, we have illustrated that Gallium-based alloys have low Seebeck coefficient. The role of this liquid metal is merely like soft electrical wire. We chosen this alloy mainly because the melting point of EGaInSn (11°C) is the lowest we can find in the Gallium-based alloys which means they will have the widest working range.
Round 2
Reviewer 1 Report
Authors have appropriately addressed the comments, however they need to expand on conclusions and discuss their work and future direction of their work in more detail.
Author Response
Authors have appropriately addressed the comments, however they need to expand on conclusions and discuss their work and future direction of their work in more detail.
Reply: Thanks for the reviewer's suggestion. We highly agree with the reviewer's comments. According to the reviewer's suggestion, we rewrote most of the clonclusions. In the revised manuscript, we add the results and conlusions from SEM into the conclusion part and gave more disscusions on our work. At the end of the conclusion we presented several directions of our future work. The revised manuscript looks more professional. Thanks for the reviewer's advice. We appriciate it.
Reviewer 2 Report
The Author have satisfactorily adressed most of my concerns. The aditional information presented in the revised manuscript adress the concerns I had with the origanal submission.
Therefore, I have no reason not to reccomend the manuscript for publication in Sensors.
Author Response
The Author have satisfactorily adressed most of my concerns. The aditional information presented in the revised manuscript adress the concerns I had with the origanal submission.
Therefore, I have no reason not to reccomend the manuscript for publication in Sensors
Reply: Thanks for the reviewer's work. We really appreciate it!